# Enhancing Maize Stress Tolerance and Productivity through Synergistic Application of *Bacillus velezensis* A6 and Lamiales Plant Extract, Biostimulants Suitable for Organic Farming

**DOI:** 10.3390/biology13090718

**Published:** 2024-09-12

**Authors:** María Peñas-Corte, Paula R. Bouzas, Juan Nieto del Río, Maximino Manzanera, Adoración Barros-Rodríguez, José R. Fernández-Navarro

**Affiliations:** 1Biopharma Research S.A., P. Industrial Autovía Norte, C/Montecillo S/N, La Carlota, 14100 Córdoba, Spain; mpenas@econatur.net; 2Department Statistics and Operations Research, Faculty of Pharmacy, Campus de Cartuja, University of Granada, 18071 Granada, Spain; paula@ugr.es; 3Laboratorios Econatur S.L., P. Industrial Autovía Norte, C/Montecillo S/N, La Carlota, 14100 Córdoba, Spain; j.nieto@econatur.net; 4Institute for Water Research and Department of Microbiology, University of Granada, Edificio Fray Luis de Granada, C/Ramón y Cajal 4, Ronda, 18003 Granada, Spain; manzanera@ugr.es; 5VitaNtech Biotechnology S.L., CL Yegen No. 3-5 °C, 18003 Granada, Spain; e.dorysbr@go.ugr.es

**Keywords:** *Bacillus velezensis*, botanical biostimulants, maize productivity, plant growth-promoting rhizobacteria (PGPR), stress tolerance, sustainable agriculture, organic farming

## Abstract

**Simple Summary:**

Maize is a vital global crop used for food and industrial purposes and whose growth is negatively affected by climate change. Through the use of combined microbial and plant-derived biostimulants, we conducted an experimental field trial on the beneficial effect it exerts on maize, in terms of improving its productivity and resistance to pathogenic toxins and adverse conditions, such as drought. This synergistic approach promises to be a valuable contribution to the field of sustainable agronomy.

**Abstract:**

Maize, a globally significant cereal, is increasingly cultivated under challenging environmental conditions, necessitating innovations in sustainable agriculture. This study evaluates the synergistic effects of a novel technique combining a *Bacillus velezensis* A6 strain with a plant extract from the Lamiales order on maize growth and stress resilience. Employing a pilot field trial, this study was conducted on the “La Añoreta” experimental farm of the ECONATUR group, where various biostimulant treatments, including bacterial and plant extract applications, were tested against a control group. The treatments were applied during key vegetative growth stages (V10-Tenth-Leaf, VT-Tassel, R1-Silking) and monitored for effects on plant height, biomass, and fumonisin content. The results suggest that the combined treatment of *Bacillus velezensis* A6 and the plant extract increases maize height (32.87%) and yield (62.93%) and also reduces fumonisin concentrations, improving its resistance to stress, compared to the control and other treatments. This study highlights the potential of microbial and botanical biostimulants and its novel combination for improving crop productivity and sustainability, suggesting that such synergistic combinations could play a crucial role in enhancing agricultural resilience to environmental stresses.

## 1. Introduction

Maize is a crucial global crop, providing food, animal feed and raw materials for industrial purposes, and its production is expected to double by 2050 to meet growing demands [1,2,3,4]. As a major source of calories and protein worldwide [5], it is heavily regulated in terms of contaminants, such as mycotoxins, especially in the European Union [6]. However, various abiotic stresses—including drought, salinity, and temperature extremes—are decreasing maize yields, requiring an estimated 60% increase in agricultural production by 2050 [7,8].

Abiotic stresses trigger diverse biochemical and physiological responses in plants, depending on the stage of growth and the specific species affected [9,10,11]. These responses aim to enhance survival under adverse conditions by altering leaf orientation, slowing growth, and modifying transpiration and nutrient distribution [12,13,14,15,16,17]. For instance, in response to drought, plants reduce water loss by closing stomata and expand their root systems to access deeper water reserves [9,18]. Photosynthesis is often reduced to prevent cellular damage from excess energy [19].

On a biochemical level, plants accumulate osmoprotectants, such as proline, glycine betaine, and soluble sugars, which stabilize cellular structures and help maintain osmotic balance during water or salt stress [13]. To combat oxidative stress, plants produce antioxidants, including superoxide dismutase (SOD), catalase (CAT), and ascorbate peroxidase (APX), which neutralize reactive oxygen species (ROS) generated during stress [20]. Additionally, phenolic compounds and flavonoids serve as antioxidants, protecting plants from UV damage and other environmental stresses [9].

At the molecular level, plants respond to abiotic stress by modulating gene expression. For example, the Dehydration Responsive Element Binding (DREB) gene family is activated during dehydration, regulating genes that protect cells and manage water content [17]. Heat shock proteins (HSPs) also play a crucial role in protein folding under stress conditions [13]. Furthermore, hormonal signaling, particularly involving abscisic acid (ABA), ethylene, jasmonic acid (JA), and salicylic acid (SA), is vital in managing plant responses to stress. ABA is particularly important for drought tolerance, as it controls stomatal closure and the expression of water-stress-related genes [21].

Biostimulants—substances or microorganisms applied to plants or soils—are increasingly used in agriculture to improve plant nutrition and stress tolerance, regardless of their nutrient content [22]. These products are crucial for enhancing crop resilience, increasing yields, and reducing dependence on synthetic inputs [23,24].

As environmental concerns and the high costs of synthetic products rise, interest is growing in sustainable, organic alternatives like plant growth-promoting rhizobacteria (PGPR), which enhance plant growth and stress tolerance through mechanisms such as hormone modulation and improved nutrient uptake [25,26]. For example, certain PGPR strains produce ABA and indole acetic acid or regulate plant ethylene levels by producing 1-aminocyclopropane-1-carboxylic acid (ACC) deaminase [27,28]. PGPR can also fix atmospheric nitrogen, mobilize soil phosphate, and produce siderophores that chelate iron [28].

The beneficial effects of plant extracts on plant growth are well documented [29], and recent research highlights the role of specific microorganisms in improving water stress tolerance [30]. The purpose of this study is to agronomically evaluate the biostimulant benefits of microorganisms and botanical extracts on the development of the maize crop and its resilience to stress. It was carried out through a pilot test under field conditions, to explore the synergistic potential of a novel, never-before-evaluated combination of *Bacillus velenzensis* A6 with a plant extract derived from a species within the Lamiales order to improve agricultural productivity and sustainability.

The findings support the continued exploration and optimization of biostimulant formulations, potentially leading to more resilient agricultural practices that align with sustainable, environmentally conscious food production systems.

## 2. Materials and Methods

### 2.1. Biostimulant Treatments

Biostimulant treatments employed included *Bacillus velezensis* A6 (hereafter referred to as A6), a strain belonging to the Operational Group of *Bacillus amyloliquefaciens* (OGBa), identified, characterized, and deposited at an International Type Culture Collection (Acc. No. SAMN40153660) by VitaNtech Biotechnology SL (Granada, Spain), functions as a plant growth-promoting rhizobacteria (PGPR) and provides a fungal pathogen control [31]. Production was conducted at the VitaNtech Biotechnology SL facilities, using 2 L flasks filled with TSB medium, cultivated in batch mode at 180 rpm and 30 °C [31], until reaching a final concentration of 10^8^ CFU/mL.

Another treatment, coded as A6LIS, aimed to investigate whether endogenous compounds produced by A6 and released into the medium upon cell wall disruption could perform a biostimulant function. For this purpose, A6 cultures were autoclaved for 20 min at 1 atm of pressure and 121 °C. To verify the sterilizing effect, 100 µL from each autoclaved batch was plated on TSA plates, confirming the loss of growth capability.

The combined treatment, coded as A6 + BS, involved A6 with a commercial biostimulant based on botanical extracts developed by the ECONATUR group. This botanical extract formulation is officially registered as a biostimulant product and authorized for use in organic production in accordance with European regulations. The plant extract is derived from agricultural by-products, rich in polyphenols and hydroxybenzoic acids with strong antioxidant capabilities through the enzymatic activation of SOD (superoxide dismutase), GR (glutathione reductase) and APX (ascorbate peroxidase) and complexing properties that help plants cope with abiotic stress [20,32]. This is facilitated by its phenolic content, allowing plants to allocate energy to their primary physiological functions rather than diverting it entirely to the antioxidant system to mitigate reactive oxygen species that induce senescence and cell damage [32,33].

As a positive control in the pilot study, a commercially known strain with PGPR and fungicidal properties and another member of the OGBa (henceforth coded as CR for Commercial Reference), *Bacillus amyloliquefaciens* D747 subsp. *Plantarum*, was included. This strain was provided by the University of Granada.

### 2.2. Experimental Pilot Field Trial Design

The pilot experiment was conducted from March to August 2021 on the experimental farm “La Añoreta” of the ECONATUR group located in Santaella (37°58′28′′ N, 4°82′97′′ W), Córdoba, spanning 504 m^2^ (Figure 1). The trial area was divided into blocks with four replications. Margin strips were maintained to avoid edge effects.

The trial was conducted on elementary plots measuring 21 m^2^ (7 × 3 m), with a row spacing of 75 cm and 50 cm between plants. This arrangement resulted in a population of 56 plants per plot, which is equivalent to 26,666 plants per hectare. All trial plots received a basal and top-dressing management regimen to address nutritional deficiencies. The basal application consisted of a compost-based fertilizer, Econatur Naturgan 545 at 1000 kg/ha (50-40-50 NPK), and the top-dressing consisted of a liquid organic fertilizer, Econatur Fulvital 60 L/ha (supplying 110 units of N), both of which are commercial products, suitable for organic farming, from Laboratorios Econatur (Córdoba, Spain).

The dent corn hybrid variety DKC6092, obtained from DEKALB, was sown at a rate of 3 seeds per hole and thinned post-germination. Experimental treatments were applied during the vegetative development from the tenth-leaf stage (V10) onwards [34]. The soil texture was sandy clay, typical on this area. Accumulated precipitation during the trial period was 50.8 L/m^2^, with average temperatures ranging from 13.9 °C to 31.8 °C and relative humidity between 32.1% and 94.7%.

The treatments applied in this pilot study were T1-control (untreated control), T2-A6, T3-A6 + BS, T4-A6LIS, and T5-CR. Three applications of each treatment were made at 15–20-day intervals. The first application occurred on the eighth week post-sowing, coinciding with the 10-true-leaf stage (V10). The second application occurred on week 11, when the maize was in the tasseling stage (VT), and the final application occurred on week 13, coinciding with the silking stage (R1). Treatments were applied at rate of 5 mL per liter of water used via automated drip irrigation starting from a treatment-specific application tank. Irrigation was reduced to simulate water stress conditions, coinciding with the fact that it was a year of extreme drought. Irrigations were only carried out in the treatment application mixture, assuming a total volume of water provided of 12 L per plot and application.

### 2.3. Growth, Quality, and Yield/Production Parameter Evaluation

Growth parameters were assessed by measuring average corn height, according to the methodology described by the International Maize and Wheat Improvement Center [35], measuring from the base of the plant to the start of the male inflorescence division using a tape measure, from eight randomly selected plants from the central two lines per treatment.

As a quality parameter within this study, both cob and grain weights were meticulously evaluated to gauge crop yield and quality. For each replication within the experiment, samples comprising three cobs (*n* = 12 per treatment) were collected during the physiological maturity stage of the crop. Each cob was accurately weighed to ensure exact data capture. Following this, the cobs were manually threshed, and the grains obtained from this process were also weighed. The meticulous weighing of both cobs and grains enabled the calculation of the grain–cob ratio. Thus, this parameter not only reflects the direct yield potential of the crop but also provides insights into the physiological efficiency of resource utilization by the plants [36].

For the analysis of average yield and production, a 1 m^2^ wooden quadrant was randomly placed in the central two lines of each plot, ensuring it was positioned away from the plot edges to avoid edge effects. The cobs from this defined area were then harvested, adhering to the protocol outlined by [37], albeit with minor adaptations tailored to our specific conditions. Subsequently, the yield data were extrapolated from grams per square meter (g/m^2^) to kilograms per hectare (kg/ha) using an established conversion factor. Furthermore, the weight of 1000 grains from these collected cobs was precisely measured, and the moisture content of these grains was assessed. These measurements allowed for the determination of the specific weight and total kernel weight (TKW). These metrics were obtained using standardized methodologies. For specific weight, the method involved filling a standardized container with a known volume of grain, which is then weighed to calculate the density [37]. For TKW, the grains were counted and weighed, and the average weight per 1000 grains was calculated [38].

### 2.4. Fumonisin Determination and Quantification

Sample preparation, fumonisin extraction, and analysis via LC–MS/MS (Agilent Technologies, Waldbronn, Germany) were also conducted by Biopharma Research. The extraction of fumonisins was carried out according to the method proposed by [39], with slight modifications. A total of 20 g of ground maize grain was weighed into a blender jar, to which 2 g of NaCl and 100 mL of extraction solvent (1:5) were added. The mixture was homogenized using an Ultra Turrax blender (IKA, Staufen, Germany) at low speed for 1 min to ensure thorough mixing with the solvent, followed by 2 min at high speed. The mixture was then filtered through Whatman No. 4 filter paper and collected in a 100 mL Erlenmeyer flask. Subsequently, 10 mL of the clear filtrate was transferred to a flask, and 40 mL of PBS solution was added. Then, 10 mL of the filtered extract was passed through an immunoaffinity column containing a gel bed with toxin-specific antibodies coupled to the gel particles (StarLine™ Immunoaffinity columns, Romer Labs Division Holding GmbH, Getzersdorf, Austria). Finally, the sample was analyzed by LC–MS/MS (Agilent Infinity 1260 HPLC coupled to a triple-quadrupole LCMS/MS QqQ 6470 Jet Stream Agilent mass spectrometer, Agilent Technologies, Waldbronn, Germany).

### 2.5. Statistical Analysis

The experimental data obtained were processed by IBM SPSS STATISTICS v. 28.0.0.0 (190) statistical analysis software. A descriptive study of the variables has been conducted and the intuitive conclusions have been compared with the results of non-parametric hypothesis tests in some cases. Clearly, the median of the variables is an especially interesting descriptive statistic. The significance level for all the tests performed is *α* = 0.05 as usual.

## 3. Results

### 3.1. Maize Growth Development

Plant height is a widely recognized indicator of overall plant vigor and health status. Taller plants generally demonstrate a robust ability to utilize available nutrients and water efficiently, which contributes to their vigorous growth. In agricultural research and practice, measuring plant height is used as a straightforward, non-destructive method to assess the general health and developmental progress of crops. This metric is particularly valuable as it often correlates with other vital growth parameters, such as biomass accumulation and yield potential. The height of the maize plant was observed in eight plants for each treatment. Table 1 shows the basic descriptive statistics.

The box plot (Figure 2) shows distinct differences in plant height across treatments (the Kruskal–Wallis test was checked, and its conclusion coincides *p*_height_ < 0.001). On the other hand, the height appears to be similar for the control data, A6LIS treatment, and CR treatment. In addition, a similar effect was observed for A6 and A6 + BS treatments.

Both the application of A6 + BS and A6 resulted in a greater total plant height compared to the control. For both experimental treatments, heights of 239.5 and 236.13 cm were achieved, representing increases of 32.87% and 31.00%, respectively, compared to the control, where an average plant height of 180.25 cm was recorded. Additionally, both treatments also differed from the CR treatment which recorded an average height of 192.5 cm. The A6LIS treatment did not improve plant height compared to the control. The best height increase was obtained after applying the combined *Bacillus velezensis* A6 and plant extract treatment.

### 3.2. Corncob Weight, Grain Weight, and Grain–Cob Ratio

The grain–cob corn ratio serves as a crucial indicator of the energy efficiency with which the plant allocates resources towards grain production versus cob development. A higher grain–cob ratio suggests more efficient resource use in producing economically valuable grain rather than non-marketable cob biomass. The study of the corncob weight (g), grain weight (g), and grain–cob ratio is based on 12 data points collected in each of the treatments. The basic descriptive statistics are shown in Table 2.

The box-and-whisker plots (Figure 3) suggest that there is not clear difference between the corncob weight among the treatments, as well as the grain weight or the grain–cob ratio. The Kruskal–Wallis tests support those conclusions (*p*_cob weight_ = 0.103, *p*_grain weight_ = 0.188, and *p*_grain–cob ratio_ = 0.867).

On the other hand, the different variability of the grain–cob ratio among the treatments seems notable. The dispersion in the control treatment is much bigger (variation coefficient, VC_control_ = 0.401) than that of the dispersion in any of the other treatments, for which variability is similar (VC_A6_ = 0.055, VC_A6+BS_ = 0.054, VC_A6LIS_ = 0.094, VC_CR_ = 0.148). The Levene test supports this finding as *p* < 0.001. The rate of the grain weight over the cob weight is more homogeneous in the treated crop and more heterogeneous in the control crop. Therefore, the means and medians of the grain–cob ratio are more reliable for any treatment than for the control group; this is a very desirable feature (Figure 4).

Nevertheless, the average cob weight obtained in plots where the A6 + BS and A6 treatment were applied was 169.25 and 162.33 g, respectively (Table 2). This represents an increase of 12.58% and 7.98%, compared to the control group, which recorded an average cob weight of 150.33 g. Only a 3% increase in cob weight was obtained with CR treatment (155.36 g). The most unfavorable result was shown by the A6LIS treatment, which reduced the cob weight by 17.29% compared to the control. Regarding grain quality, measured as average weight, A6 + BS stands out from the rest of the theses evaluated in the trial as seen in Table 2, representing an increase of 12.95% compared to the control. This is followed by the treatment A6 and CR, with a higher total grain weight of 132.67 and 124.86 g. Again, a lower average grain weight is obtained with A6LIS, showing a 19% reduction compared to the control.

Despite not having obtained clear differences in terms of cob weight, grain weight, or grain–cob ratio between all the treatments, after applying both *Bacillus velezensis* A6 and its combination with the biostimulant plant extract, a much more homogeneous grain–cob ratio was obtained (Figure 4).

### 3.3. Average Maize Yield

The specific weight of maize grain of corn is an important quality attribute as it indicates the specific weight of the grain, which is crucial for assessing grain filling and maturity. A higher specific weight generally suggests better kernel development and potentially higher starch content, which are desirable traits for both marketability and processing quality. The TKW is equally critical as it provides a measure of the average weight of the grains, serving as an indicator of seed size and uniformity, which are key factors in assessing the genetic potential and health of the crop. The production (kg/ha), moisture (%), specific weight of maize grain (kg/hL), and TKW (g) are studied using the four data points per treatment. Table 3 shows the descriptive statistics, and Figure 5 shows the corresponding box-and-whisker plots.

The graphics in Figure 5 let us surmise that production, moisture, and TKW are different among the treatments. Having this sample size (n = 4), it would not be appropriate to perform any hypothesis test; in any case, it is interesting that even in this scenario, the tests coincide with the presented idea (Kruskal–Wallis *p*-values are *p*_production_ = 0.008, *p*_moisture_ = 0.039, and *p*_TKW_ = 0.045). On the other hand, the specific weight appears to be similar in the different treatments.

Regarding the average yield obtained from maize and the application of A6 + BS, we observe at harvest an increase of 62.93% in production compared to the untreated control, resulting in a yield of 4183.3 kg of grain per hectare compared to 2567.4 kg. What is also noteworthy is that the increase in yield obtained with A6, 3642.9 kg/ha, has a value very similar to that obtained with CR treatment, 3246.7 kg/ha. Both treatments represent an increase of 41.89% and 26.45%, respectively, compared to the untreated control. After applying A6LIS, the average production is reduced by 32.69% compared to the control.

The treatments seem to form a couple of groups of similar production: the control and A6LIS treatments and the other three. Besides this, the observed production is always smaller for the A6LIS treatment than for the A6, A6 + BS, and CR treatments.

Concerning the moisture, which serves as an indicator of better water retention and management under drought conditions, it is notable that it is much smaller for the control data in comparison with any other treatment. Furthermore, the moisture for A6, A6 + BS, A6LIS, and CR seems to be similar. On the other hand, the difference in variability draws attention as it seems smaller for the control data (even the Levene test does not back up this idea, as *p* = 0.099).

Even the median specific weight seems to be similar for the different treatments, it is also notable that the variability is bigger for the control group and the A6LIS treatment (VC_control_ = 0.071, VC_A6LIS_ = 0.097) than for the other groups (VC_A6_ = 0.031, VC_A6 + BS_ = 0.035, VC_CR_ = 0.032). Once again, smaller variability is a good feature, as seen in A6, A6 + BS, and CR.

The TKW was smaller for the control data, while the rest of the treatments showed a similar TKW. In this respect, the variability shows again a remarkable difference between treatments (the Levene test supports the idea, as *p* = 0.004). The control data are clearly more heterogenous (VC_control_ = 0.075) than the others (VC_A6_ = 0.030, VC_A6 + BS_ = 0.02, VC_A6LIS_ = 0.0042, VC_CR_ = 0.023). The control group is formed by much more diverse plants. The A6 + BS treatment shows the smallest variability, which is an important fact.

After measuring all variables related to crop yield, the combined application of *Bacillus velezensis* A6 and the biostimulant plant extract led to significantly higher corn production compared to other treatments.

### 3.4. Fumonisin Content Reduction

The descriptive statistics of the fumonisin content for each treatment is presented in Table 4. Figure 6 presents a box-and-whisker plot, and it appears that there is no significant difference between the treatments (the Kruskal–Wallis test supports it, as *p* = 0.531). In any case, the A6 treatment could have less variability than the other treatments (see the variation coefficients in Table 4). In fact, the variation coefficient is the smallest of all the others. This is an important feature to explore as it means that the fumonisins in the plants of this treatment are more homogenous; therefore, the mean is more representative. It is essential to be confident that the fumonisin variability is small as a control quality indicator.

The *Bacillus velezensis* A6 treatment exhibits the lowest fumonisin content, suggesting an inhibitory effect on *Fusarium* spp., which cause mycotoxin contamination.

## 4. Discussion

This study investigated the role of biostimulants in improving maize resilience and productivity. The focus was on the novel synergy between *Bacillus velezensis* A6 and plant-derived extract under field drought conditions. The findings of this research highlight the potential of synergistic biostimulants, also suitable for organic agriculture, to enhance crop yield, plant height, and fumonisin management; these are key indicators of healthy and vigorous crop development.

It is possible to improve the absorption mechanisms of nutrients and their efficiency, in addition to boosting tolerance against abiotic stress, which are all environmental factors that alter the physiological processes of plants, thus affecting their development [22].

The results reveal that treatment with *Bacillus velezensis* A6 led to an increase in both cob and grain weights and also in the quality of marketable grains compared to the control (Table 2, Figure 3 and Figure 4). These findings are in concordance with previous studies suggesting that the application of this PGPR A6 strain can improve plant growth parameters in garlic, tomatoes, lettuce, and onions, potentially by enhancing nutrient uptake and hormonal modulation, as has been recently published [31], and coinciding with other *B. velezensis* strains [40,41,42].

The phenolic compounds stand out for their application as antioxidants and in the process of metal ion chelation, direct action on natural phytohormones, and stabilization of ascorbic acid, among others [43]. As demonstrated in our previous studies, the content of polyphenols in the plant-derived biostimulant plays a crucial role due to their high antioxidant capacity. These compounds neutralize free radicals, enable plants to scavenge reactive oxygen species (ROS) from tissues, and inhibit lipid peroxidation [33]. This has been shown to be achieved through the induction of enzyme activities such as catalases, peroxidases, polyphenol oxidases, and superoxide dismutase [43,44,45], so they prevent oxidative stress in plants, which is often triggered by abiotic stress factors such as drought.

The combined application of *Bacillus velezensis* A6 and plant-based biostimulant, rich in polyphenols and hydroxybenzoic acids, resulted in an important increase in median plant height (Table 1) and production (Table 3 and Figure 5) values compared to the control. The biostimulant benefit of plant extracts on photosynthetic activity and secondary metabolism is widely studied and closely related to improved vegetative growth [46,47,48,49]. This height advantage correlates with the roles of *Bacillus* spp. in promoting plant growth and enhancing nutrient uptake, as well as the known effects of certain plant extracts in stimulating plant growth [33,50]. This alignment with the project’s objectives of augmenting maize development and resilience to environmental stress underscores the tangible agronomic benefits of biostimulant treatments.

By combining both microbial and plant-based biostimulants, on which there are few scientific works [51], this study aimed to address the challenges faced by crops, such as environmental stress and nutrient deficiency, providing a comprehensive response at both systemic and nutritional levels. Antioxidants derived from phenolic compounds in the plant extracts inhibit the degradation of organic acids [52], while *Bacillus velezensis* A6 utilizes these compounds to enhance its metabolism and activities as a plant growth-promoting rhizobacterium (PGPR) [53]. This combination facilitates improved water stress resistance through stomatal closure and enhanced root development, leading to better uptake of nutrients that *Bacillus velezensis* A6 mobilizes and makes available for a long time in the soil [54,55].

Chaudhary et al. reported an improvement in maize growth through the combined application of *Bacillus* spp. and zeolite, a mineral widely used as a soil enhancer that has indirect biostimulant effects on plants as it improves nutrient use efficiency. Specifically, they achieved a 29.80% increase in corn productivity and a significant improvement in plant height, along with enhanced levels of antioxidant enzymes, showing an improvement in stress management [56]. A similar study on tomatoes conducted by Shah et al. demonstrated that the combined application of zinc sulfide nanoparticles biosynthesized from an aqueous extract of *Jacaranda mimosifolia*, *Acinetobacter pittii*, and *Bacillus velezensis* improved nutrition and tomato yield, as well as reduced ROS damage caused by *Rhizoctonia solani* incidence [57].

Contrarily, the A6LIS treatment, which utilized lysed *Bacillus velezensis* A6 cells, did not deliver comparable benefits, suggesting the importance of live microbial populations in exerting biostimulant effects. This finding is supported by research indicating the critical role of live bacteria in stimulating plant growth and drought resistance [25,26].

This study also provided insights into fumonisin content, where the A6 treatment showed a reduction in fumonisins relative to the control. This suggests an inhibitory effect of *Bacillus velezensis* A6 on Fusarium species, which is critical given stringent EU regulations on mycotoxin content in maize, with 4000 ppb or µg/kg being the maximum allowable limit for fumonisins [6]. However, the CR treatment, despite incorporating a commercially recognized PGPR strain, did not demonstrate superiority in reducing fumonisins, indicating that not all PGPR strains may be equally effective against *Fusarium* spp. or that the response may be context-dependent [58,59].

It is important to note that several unforeseen issues arose during the execution of the pilot trial, preventing the collection of a minimum sample size for evaluations for each treatment. These included the lines defined to avoid the edge effect, which required the discarding of part of the plants from all treatments. Due to the nature of the soil of the trial area along with the scarcity of water, complete plant extraction was not possible as the roots were deeply anchored, necessitating in situ measurements on the field as accurately as possible. In several plot areas, weed growth competed directly with the maize plants due to the absence of phytosanitary treatments, which were omitted to maintain compliance with organic production standards.

Considering the complexities introduced by variable environmental conditions, the results highlight the need for more rigorous field trials and larger sample sizes to confirm the observed trends and explore the underlying mechanisms through which biostimulants affect crop resilience and fumonisin accumulation. This research could have profound implications for sustainable agriculture, especially in the face of climate change challenges that threaten global food security.

## 5. Conclusions

This pioneering study demonstrates that the synergistic combination of *Bacillus velezensis* A6 and extracts from Lamiales order plants enhances maize resilience and productivity under stress conditions, drought, and fungal attack. Our findings reveal that this biological interaction not only increases maize height and yield but also effectively reduces fumonisin concentrations, thus contributing to safer and more sustainable agriculture. These results highlight the potential of microbial and botanical biostimulants as innovative strategies for bolstering sustainability in global agricultural production in the face of climate change challenges and increasing nutritional demands. This study lays the groundwork for future research that could optimize and expand the use of these biostimulants across various agronomic and geographic conditions, redefining agricultural productivity paradigms in the modern era.

## Figures and Tables

**Figure 1 biology-13-00718-f001:**
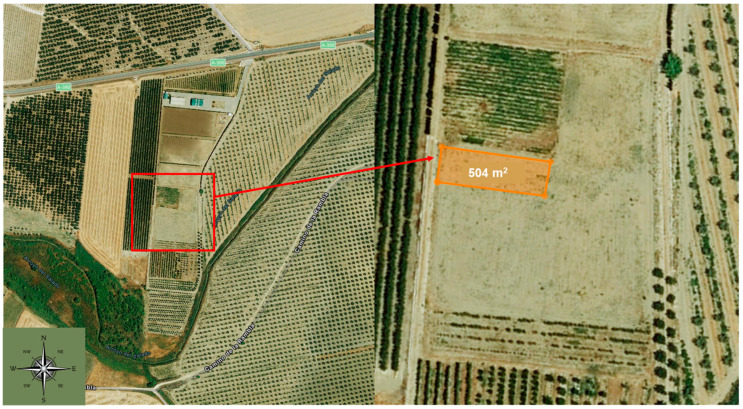
Aerial photograph of part of the experimental farm “La Añoreta”. Enlarged and marked in orange is the trial area where the experimental pilot study was carried out.

**Figure 2 biology-13-00718-f002:**
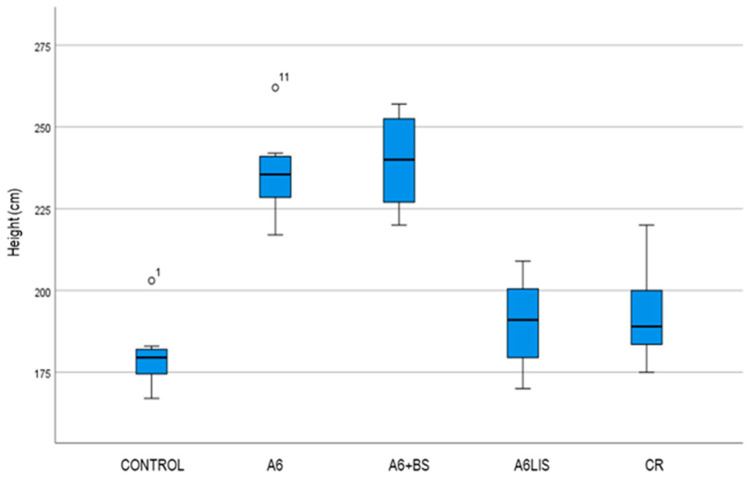
Box plot depicting the distribution of height across different treatment groups: control, A6, A6 + BS, A6LIS, and CR. The boxes represent the interquartile range (IQR), the line inside each box indicates the median, and the whiskers extend to the most extreme data points that are not considered outliers. The outliers are represented by individual points above the whiskers. This visualization aids in comparing the central tendency and variability of height across treatments.

**Figure 3 biology-13-00718-f003:**
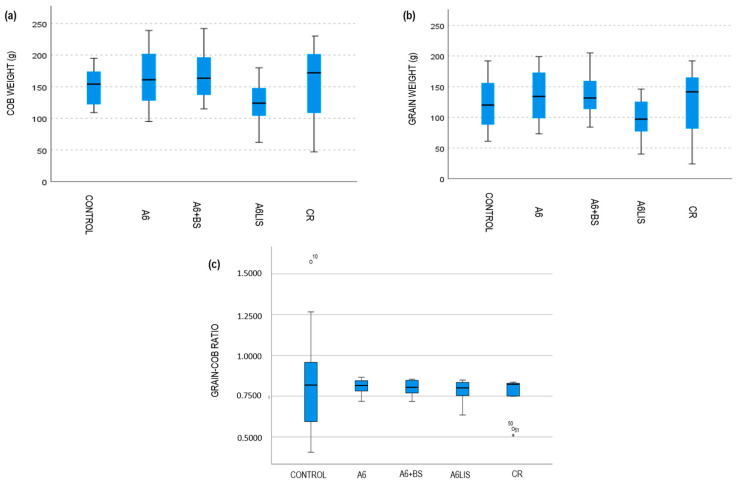
Panel of box plots displaying comparative weight data across different treatment groups. (**a**) Cob weight (g) across various treatments showing variability in weight distribution with each box representing the interquartile range and whiskers extending to the outermost data points not considered outliers. (**b**) Grain weight (g) among different groups indicating differences in grain mass with median lines visible within each box. (**c**) Grain–cob ratio presented for each treatment, highlighting significant disparities in ratios. Outliers are marked with symbols outside the whiskers. These plots facilitate the assessment of treatment effects on cob and grain weights and their ratios.

**Figure 4 biology-13-00718-f004:**
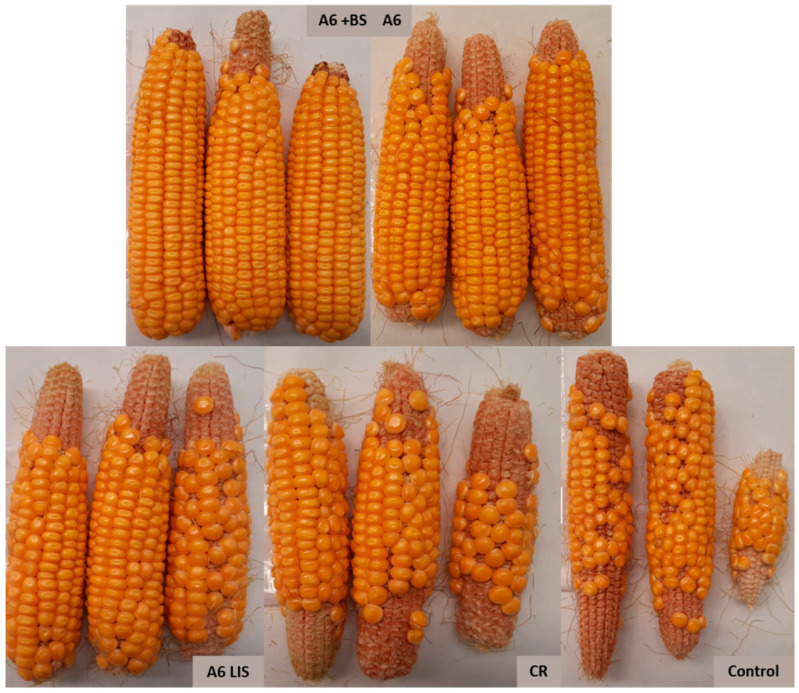
Corncobs collected during the maturation stage. The homogeneity in grain formation presented by treatments A6 + BS and A6 stands out, coinciding with the variability coefficient obtained in the determination of the grain–cob ratio.

**Figure 5 biology-13-00718-f005:**
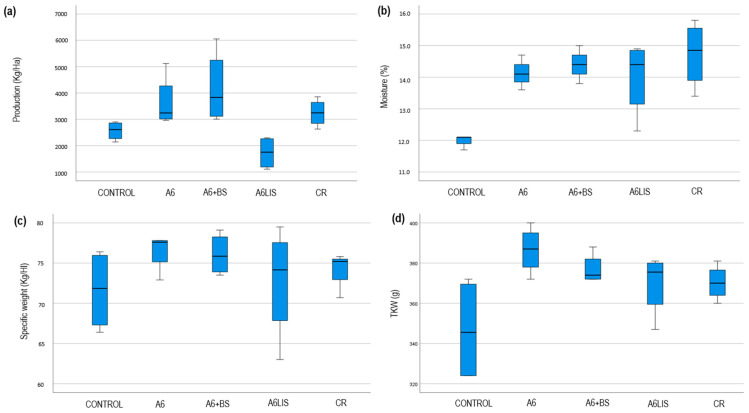
Panel of box plots displaying comparative yield data across different treatment groups. (**a**) Production (kg/ha) across various treatments showing variability in production levels, with each box representing the interquartile range and whiskers extending to the outermost data points not considered outliers. (**b**) Moisture (%) among different groups indicating differences in moisture content with median lines visible within each box. (**c**) Specific weight (kg/hL) presented for each treatment, highlighting disparities in density measurements. (**d**) TKW (g) across treatments, illustrating the commercial output variability. These plots facilitate the assessment of treatment effects on production, moisture content, specific weight, and marketable weight.

**Figure 6 biology-13-00718-f006:**
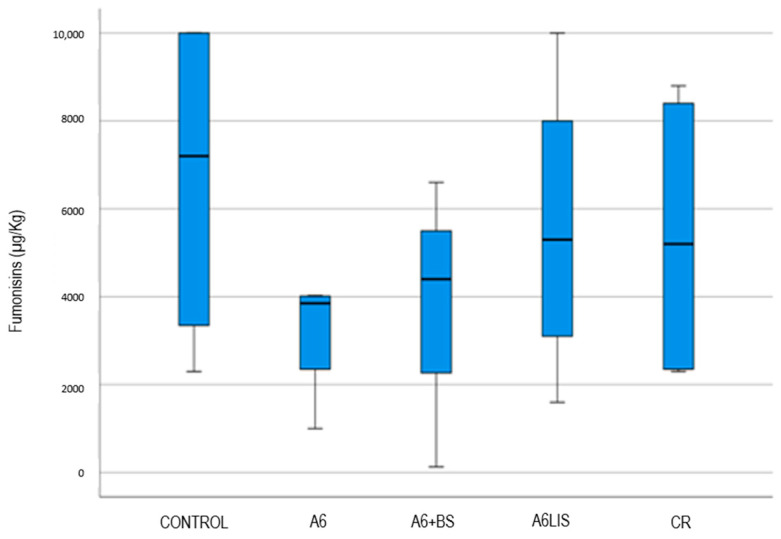
Box plot showing the distribution of fumonisin levels (µg/kg) across different agricultural treatment groups: control, A6, A6 + BS, A6LIS, and CR. The boxes represent the interquartile range (IQR) where the lower and upper quartiles are marked by the bottom and top of the box, respectively. The line within each box represents the median value, while the whiskers extend to the furthest data points not considered outliers. The A6 treatment group exhibits the lowest fumonisin content among the groups, indicating its effectiveness in minimizing fumonisin levels in crops. This visualization helps to evaluate the impact of each treatment on fumonisin contamination levels in crops.

**Table 1 biology-13-00718-t001:** Basic descriptive statistics for the height variable for treatments: *Bacillus velezensis* A6 (A6), *Bacillus velezensis* A6 + biostimulant plant extract (A6 + BS), lysed *Bacillus velezensis* A6 (A6LIS), and *Bacillus amyloliquefaciens* D747 subsp. *Plantarum* (CR). Eight plants were taken for each measurement per treatment.

	Height (cm)			
Treatments	Mean	Median	Standard Deviation	Variation Coefficient
Control	180.250	179.500	10.512	0.058
A6	236.130	235.500	13.196	0.055
A6 + BS	239.500	240.000	14.412	0.060
A6LIS	190.130	191.000	13.389	0.070
CR	192.500	189.000	14.081	0.073

**Table 2 biology-13-00718-t002:** Basic descriptive statistics for the variables corncob weight, grain weight, and grain–cob ratio for each treatment.

Treatments	Variable	Mean	Median	Standard Deviation	Variation Coefficient
Control	Cob weight	150.330	154.000	29.203	0.194
	Grain weight	121.580	120.000	40.603	0.333
	Grain–cob ratio	0.838	0.818	0.336	0.401
A6	Cob weight	162.330	161.000	46.257	0.284
	Grain weight	132.670	134.000	42.907	0.323
	Grain–cob ratio	0.808	0.816	0.045	0.055
A6 + BS	Cob weight	169.250	163.500	42.180	0.249
	Grain weight	137.330	131.500	39.213	0.285
	Grain–cob ratio	0.805	0.804	0.044	0.054
A6LIS	Cob weight	124.330	124.000	31.586	0.173
	Grain weight	98.080	97.000	30.435	0.310
	Grain–cob ratio	0.777	0.801	0.073	0.094
CR	Cob weight	153.670	172.000	62.144	0.404
	Grain weight	122.830	141.500	56.809	0.462
	Grain–cob ratio	0.764	0.823	0.113	0.148

**Table 3 biology-13-00718-t003:** Basic descriptive statistics for the variables specific weight of maize grain, moisture, TKW, and production for each treatment.

Treatments	Variable	Units	Mean	Median	Standard Deviation	Variation Coefficient
Control	Specific weight	kg/hL	71.625	71.850	5.061	0.071
	Moisture	%	12.000	12.100	0.200	0.017
	TKW	g	346.750	345.500	26.349	0.076
	Production	kg/ha	2567.381	2612.143	357.056	0.139
A6	Specific weight	kg/hL	76.475	77.600	2.390	0.031
	Moisture	%	14.125	14.100	0.450	0.032
	TKW	g	386.500	387.000	11.705	0.030
	Production	kg/ha	3642.857	3242.857	1006.203	0.276
A6 + BS	Specific weight	kg/hL	76.075	75.850	2.626	0.035
	Moisture	%	14.400	14.400	0.489	0.034
	TKW	g	377.000	374.000	7.572	0.020
	Production	kg/ha	4183.333	383.333	1397.583	0.334
A6LIS	Specific weight	kg/hL	72.700	74.150	7.041	0.097
	Moisture	%	14.000	14.400	1.202	0.086
	TKW	g	369.750	375.500	15.650	0.042
	Production	kg/ha	1727.976	1753.571	623.705	0.361
CR	Specific weight	kg/hL	74.225	75.200	2.367	0.032
	Moisture	%	14.725	14.850	1.056	0.072
	TKW	g	370.250	370.000	8.732	0.024
	Production	kg/ha	3246.667	3248.095	521.119	0.161

**Table 4 biology-13-00718-t004:** Basic statistics for the variable fumonisins for each treatment.

	Fumonisins (µg/Kg)			
Treatments	Mean	Median	Standard Deviation	Variation Coefficient
Control	6675.000	7200.000	3933.934	0.589
A6	3182.500	3850.000	1462.609	0.460
A6 + BS	3882.500	4400.000	2708.116	0.698
A6LIS	5550.000	5300.000	3488.553	0.629
CR	5375.000	5200.000	3508.442	0.653

## Data Availability

The original contributions presented in this study are included in this article; further inquiries can be directed to the corresponding author.

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
