# Peer review of "Enhancing Maize Stress Tolerance and Productivity through Synergistic Application of Bacillus velezensis A6 and Lamiales Plant Extract, Biostimulants Suitable for Organic Farming"

_biology, 2024, doi:10.3390/biology13090718_

Round 1

Reviewer 1 Report

Comments and Suggestions for Authors

Submission ID: biology-3139716

Title of the manuscript: "Enhancing maize stress tolerance and productivity through synergistic application of Bacillus velezensis A6 and Lamiales plant extract, biostimulants suitable for organic farming".

This study evaluated the impact Bacillus sp. A6 and Lamiales plant extract on maize development under stress for sustainable agricultural. The subject is very interesting, and the methodology used is adequate for the objectives of the study. The results are of interest and support the conclusions. The manuscript is worth publishing in “Biology”. However, there are still some issues need to be addressed, I suggest major revision.

Specific comments

L22-27: The presentation of the key findings of experimental results should be improved and data regarding the mainly measured indicators should be presented.

L30-31: Please use specific keywords and synonyms. In addition, please arrange the keywords in alphabetical order.

L34-58: The literature is quite repetitive, thus literature should be concentrated explaining the importance of maize plant worldwide, then extend to its problem with abiotic stresses in general.

L59-62: Not sufficient. Please extend the impact of abiotic stress on plants in the light of physiological, biochemical, and molecular approaches.

L63: Why not Bioinoculants? Why did the authors choose the term Biostimulants?

L72-73: Which mechanisms? Not sufficient.

L76-83: Solid hypothesis should give a brief about the novelty of the study and give the readers more information regarding the purpose and the mechanistic used to achieve this goal, then may refer some lack in the previous study regarding some aspects.

L86: Is A6 the accession number of the bacterium in the GenBank?

L133-134: How did the authors apply the treatments (rates)? Please clarify?

L135: Physical and chemical properties of the experimental soil?

L92: What is the composition of SG medium? Please provide a reference.

L143-144: For SPSS software, the version number is mandatory information.

L199: Scientific names should be italic throughout the manuscript.

L198-206: I suggest deleting these paragraphs. Not suitable here.

Table 1: Please provide some information such as (number of replicates (n), definition of the treatments).

In the results, there is some confusion owing to the language. So, I suggest that the authors could add one sentence at the end of each paragraph to conclude the whole paragraph to make it easy for the reader.

The authors should cite the Tables and Figures in the discussion part to make the readers in touch with their results.

L179,397: Follow the journal notation in citing references. Please revise the whole manuscript.

Overall, the transition between paragraphs in the introduction needs more improvement.

L434-444: I suggest deleting this paragraph. Not suitable here, it is suitable for the conclusion section.

Regards.

Comments on the Quality of English Language

Minor editing of English language required

Author Response

1. Summary

Thank you very much for taking the time to review this manuscript. Please find the detailed responses below and the corresponding revisions/corrections in track changes in the re-submitted files.

2. Point-by-point response to Comments and Suggestions for Authors

Comments 1: L22-27: The presentation of the key findings of experimental results should be improved and data regarding the mainly measured indicators should be presented.

Response 1: Thank you for pointing this out. We agree with this comment. Therefore, we have reviewed the presentation of results. The treatments were applied during key vegetative growth stages (V10-Tenth-Leaf, VT-Tassel, R1-Silking) and monitored for effects on plant height, biomass, and fumonisins content. Results suggest that the combined treatment of Bacillus velezensis A6 and the plant extract increases maize height (32.87%) and yield (62.93%), and also reduce fumonisins concentrations, improving its resistance to stress, compared to control and other treatments. This study highlights the potential of microbial and botanical biostimulants to improve crop productivity and sustainability, suggesting that such synergistic combinations could play a crucial role in enhancing agricultural resilience to environmental stresses (L31-38).

Comment 2: L30-31: Please use specific keywords and synonyms. In addition, please arrange the keywords in alphabetical order.

Response 2: Agree. We have, accordingly modified “Bacillus velezensis, botanical biostimulants; maize productivity; plant growth-promoting rhizobacteria (pgpr); stress tolerance; sustainable agriculture; organic farming” to improve this point (L39-40)

Comment 3: L34-58: The literature is quite repetitive, thus literature should be concentrated explaining the importance of maize plant worldwide, then extend to its problem with abiotic stresses in general.

Response 3: Agree. We have, accordingly reduced in the manuscript L43-49.

Comment 4: L59-62: Not sufficient. Please extend the impact of abiotic stress on plants in the light of physiological, biochemical, and molecular approaches.

Response 4: Agree. We have, accordingly extended in the manuscript L50-71.

Comment 5: L63: Why not Bioinoculants? Why did the authors choose the term Biostimulants?

Response 5: Thank you for pointing this out. We decided this term because the definition of bioinoculant only includes microorganisms or their preparations, while biostimulant includes other substances in addition to microorganisms in its definition. In this work, the use of bacteria and plant extract is studied, so it is more appropriate to use the term biostimulant, which is officially recognized for that purpose

Comment 6: L72-73: Which mechanisms? Not sufficient.

Response 6: Agree. We have, accordingly extended in the manuscript L79-83.

Comment 7: L76-83: Solid hypothesis should give a brief about the novelty of the study and give the readers more information regarding the purpose and the mechanistic used to achieve this goal, then may refer some lack in the previous study regarding some aspects.

Response 7: Thank you for pointing this out. We highlight the novelty of the study in the manuscript L86-91. For further clarification, this work is the confirmation at field conditions of what we have observed in the recently accepted article in which it is verified that the Agricultural Protection Against Stress Index, which gave higher values to B. velezensis A6 than to B. amyloliquefaciens D747 subsp. Plantarum, is corroborated in field. In addition to the cooperative effect of a microbial biostimulant combined with one of plant originWe

Comment 8: L86: Is A6 the accession number of the bacterium in the GenBank?

Response 8: No, is the number code of the strain. It was internally codified by the authors. This strain is phylogenetically close to B. velezensis CR-502(T) with accession number AY603658. The similarity is 99.92 % with B. velezensis A6, which is privately deposited on international Type Culture Collection (Acc. No. SAMN40153660)

Comment 9: L133-134: How did the authors apply the treatments (rates)? Please clarify?

Response 9: Thank you for pointing this out. All treatments were applied at a rate of 5 ml per liter of water used for irrigation. We have included it in the manuscript L154-155

Comment 10: L135: Physical and chemical properties of the experimental soil?

Response 10: Thank you for pointing this out. During the trial design, we performed a basic analysis of the soil at Biopharma Research facilities. We obtained the following results:

Physical properties:

- texture: sand (72% w/w), silt (6% w/w) and clay (22% w/w) measured by Bouyoucos hydrometer method

Chemical properties:

- pH: 7,8 measured by Potentiometric pH meter

- Conductivity at 20ºC: 660.00 µS/cm measured by Electrical conductivity meter

- Organic matter: 0.63 % w/w measured by volumetric method (calcination)

Comment 11: L92: What is the composition of SG medium? Please provide a reference.

Response 11: Agree. We have accordingly modified in manuscript (L102-103) because it was a mistake. The medium was TSB included in: Barros-Rodríguez A, Pacheco P, Peñas-Corte M, Fernández-González AJ, Cobo-Díaz JF, Enrique-Cruz Y, et al. Comparative Study of Bacillus-Based Plant Biofertilizers: A Proposed Index. Biology. 2024 Sep;13(9):668.

Comment 12: L143-144: For SPSS software, the version number is mandatory information

Response 12: Agree, we have accordingly included in the manuscript L203-204.

Comment 13: L199: Scientific names should be italic throughout the manuscript.

Response 13: Thank you for pointing this out. We have accordingly revised the whole manuscript

Comment 14: L198-206: I suggest deleting these paragraphs. Not suitable here.

Response 14: Thank you for this suggestion. We have accordingly deleted

Comment 15: Table 1: Please provide some information such as (number of replicates (n), definition of the treatments).

Response 15: Thank you for this suggestion. We have accordingly included in manuscript

Comment 16: In the results, there is some confusion owing to the language. So, I suggest that the authors could add one sentence at the end of each paragraph to conclude the whole paragraph to make it easy for the reader.

Response 16: Thank you for this suggestion. We have accordingly included a summary sentence at the end of paragraphs.

Comment 17: The authors should cite the Tables and Figures in the discussion part to make the readers in touch with their results.

Response 17: Thank you for this suggestion. We have accordingly cited in the discussion.

Comment 18: L179,397: Follow the journal notation in citing references. Please revise the whole manuscript.

Response 18: Agree, we have revised the whole manuscript

Comment 19 Overall, the transition between paragraphs in the introduction needs more improvement.

Response 19: Thank you for pointing this out. We have accordingly revised all the introduction after your comments.

Comment 20: L434-444: I suggest deleting this paragraph. Not suitable here, it is suitable for the conclusion section.

Response 20: Thank you for this suggestion. We have accordingly deleted

3. Response to Comments on the Quality of English Language

Point 1: Minor editing of English language required

Response 1: Agree, we have accordingly revised the whole manuscript for better flow and clarity

Reviewer 2 Report

Comments and Suggestions for Authors

Comments and Suggestions for Authors

This manuscript conducted Enhancing maize stress tolerance and productivity through syn-2 ergistic application of Bacillus velezensis A6 and Lamiales plant 3 extract, biostimulants suitable for organic farming. The study was meaningful; however, the paper is interesting, but has some deficiencies or shortcomings that need to be corrected or completed.

The abstract need to be improved to highlight the theme of MS. Main results should be clearly stated in this section

Line 19. Plant extract unknown just remembered Lamiales order (The order Lamiales are an order in the asterid group of dicotyledonous flowering plants. It includes about 23,810 species, 1,059 genera, and is divided into about 25 families).

Line 23. Abbreviations should be explained in the first place they are mentioned (V10, VT, R1)

Line 30.

Keywords: Maize productivity; Biostimulants; Plant growth-promoting rhizobacteria (PGPR); stress 30 tolerance; Sustainable agriculture; Organic farming (name in lower case).

Line 144. water stress conditions ( need a more detailed explanation)

All  figures need to be improved, significant analysis were necessary in the study. Most of the results were lack of the necessary notes

 Some recent literature need to be added to comparatively discuss with the results.

Comments on the Quality of English Language

 Moderate editing of English language required.

Author Response

1. Summary

Thank you very much for taking the time to review this manuscript. Please find the detailed responses below and the corresponding revisions/corrections in track changes in the re-submitted manuscript.

2. Point-by-point response to Comments and Suggestions for Authors

Comment 1: The abstract need to be improved to highlight the theme of MS. Main results should be clearly stated in this section

Response 1: Thank you for pointing this out. We agree with this comment. Therefore, we have reviewed the abstract to highlight our study and the main results.

Comment 2: Line 19. Plant extract unknown just remembered Lamiales order (The order Lamiales are an order in the asterid group of dicotyledonous flowering plants. It includes about 23,810 species, 1,059 genera, and is divided into about 25 families).

Response 2: Thank you for pointing this out. We agree with this comment. This plant extract is derived from Olea europaea, it is a patented extract from Laboratorios Econatur

Comment 3: Line 23. Abbreviations should be explained in the first place they are mentioned (V10, VT, R1)

Response 3: Agree. We have accordingly clarified the abbreviations in manuscript L31-32

Comment 4: Keywords: Maize productivity; Biostimulants; Plant growth-promoting rhizobacteria (PGPR); stress 30 tolerance; Sustainable agriculture; Organic farming (name in lower case).

Response 4: Thank you for pointing this out. We have accordingly modified the keywords in manuscript L39-40

Comment 5: Line 144. water stress conditions ( need a more detailed explanation)

Response 5: Thank you for pointing this out. The maize irrigation was limited to reproduce water deficiency conditions. And also coinciding with the lack of rain during the trial experiment (only 50.8 L/m2) because it was a year of extreme drought.  Irrigations were only carried out in the treatment application mixture, assuming a total volume of water provided of 12 L per plot and application.

Comment 6: All figures need to be improved, significant analysis were necessary in the study. Most of the results were lack of the necessary notes

Response 6: Thank you for pointing this out. It is important to note that several unforeseen issues arose during the execution of the pilot trial, preventing the collection of a minimum sample size for evaluations for each treatment. These included bursting of irrigation lines due to water pressure, requiring the discarding of plants from all treatments. Due to the soil nature of the trial area, complete plant extraction was not possible as the roots were deeply anchored, necessitating in-situ measurements on the field as accurately as possible. In several plot areas, excessive weed growth directly competed with the corn plants, limiting their growth.

The interest of the topic under discussion is important enough to perform a statistical analysis of the collected data despite the above-mentioned limitations of our experiment. These limitations lead to small sample sizes (n = 12) or even very small (n = 4, 8). Therefore, a descriptive study of the variables has been conducted and the conclusions have been compared with the results of non-parametric hypothesis tests in some cases. Clearly, the median of the variables is a specially interesting descriptive statistic. The significance level for all the tests performed is α = 0.05 as usual.

Comment 7: Some recent literature need to be added to comparatively discuss with the results.

Response 7: Thank you for pointing this out. At this point, it has not been easy for all authors, after an intense search, to compile more works than those already referenced in the discussion, in which the agronomic effect on plants after applying a combination of bacteria and plant extracts has been studied. From here we can derive the novelty of this pilot study and our commitment and future perspective to continue researching this promising combination.

3. Response to Comments on the Quality of English Language

Point 1: Moderate editing of English language required

Response 1: Agree, we have accordingly revised the whole manuscript for better flow and clarity

Round 2

Reviewer 1 Report

Comments and Suggestions for Authors

This is the second time I have evaluated this manuscript. The authors addressed all my comments, and the manuscript has been noticeably improved. Many thanks for their contribution.

Comments on the Quality of English Language

Only minor editorial and stylistic corrections are required.

Reviewer 2 Report

Comments and Suggestions for Authors

Thanks for correcting the required points.